# TOPIC-BASED QUESTION GENERATION

## ABSTRACT

Asking questions is an important ability for a chatbot. This paper focuses on question generation. Although there are existing works on question generation based on a piece of descriptive text, it remains to be a very challenging problem. In the paper, we propose a new question generation problem, which also requires the input of a target topic in addition to a piece of descriptive text. The key reason for proposing the new problem is that in practical applications, we found that useful questions need to be targeted toward some relevant topics. One almost never asks a random question in a conversation. Due to the fact that given a descriptive text, it is often possible to ask many types of questions, generating a question without knowing what it is about is of limited use. To solve the problem, we propose a novel neural network that is able to generate topic-specific questions. One major advantage of this model is that it can be trained directly using a question-answering corpus without requiring any additional annotations like annotating topics in the questions or answers. Experimental results show that our model outperforms the state-of-the-art baseline.

## 1 INTRODUCTION

The goal of question generation is to generate questions according to some given information (e.g., a sentence or a paragraph). It has been applied in many scenarios, e.g., generating questions for reading comprehension (Du et al., 2017; Heilman & Smith, 2010) and generating data for large scale question-answering training (Serban et al., 2016; Duan et al., 2017). Since questioning is an important communication skill, question generation plays an important role in both general-purpose chatbot systems and goal-oriented dialogue systems. In the context of dialogue, many researchers have studied the problem (Mostafazadeh et al., 2016; Bordes & Weston, 2017). The generated questions are mainly used to start a conversation or to obtain some specific information.

Earlier approaches to question generation mainly used human-crafted rules and patterns to transform a descriptive sentence to a related question (Mitkov, 2003; Chali & Hasan, 2015). However, human-crafted rules are limited. They cannot effectively cover a large number of question generation scenarios. Deep neural networks learned by end-to-end methods can help overcome this problem. This approach has been successfully applied to many NLP tasks, e.g., neural machine translation (Bahdanau et al., 2015; Sutskever et al., 2014), summarization (Iyer et al., 2016), etc. At the same time, some training optimization studies have further guaranteed the performance and stability of end-to-end networks (Mishkin & Matas, 2016; Cogswell et al., 2016). For question generation, Serban et al. (2016) applied a neural network to the knowledge base Freebase to transduce facts to questions. Du et al. (2017) introduced an attention-based sequence learning model, which outperformed state-of-the-art rule-based systems. Two approaches were proposed by Duan et al. (2017). One is retrieval-based, and the other is generation-based. Du & Cardie (2017) also studied question-worthy sentences in reading comprehension.

Much of the existing question generation studies mainly focused on the task of generating questions from sentences, paragraph, or structured data only. In this paper, we argue that *topic-based question generation* is also very important. That is, in addition to the given sentence or paragraph, it is also useful to specify a relevant topic contained in the text. The main reason is that a sentence or paragraph often involves multiple topics or concepts that questions can be generated, only arbitrarily choose one or mixing them may be of limited use because we found that in practical applications, questions need to be targeted toward some topics related to the current conversation. One almost never asks a random question in a conversation. Generating a question without knowing what it is

about is not very useful. To solve the proposed problem, we propose a novel neural network that can generate topic-based questions. One major advantage of our model is that it can be trained directly using a question-answering corpus without requiring any additional annotations like annotating the topics in the questions or answers. In summary, this paper makes the following contributions:

- It proposes the new problem of question generation based on a given sentence and a topic (or concept) in the sentence. To our knowledge, this topic-based question generation has not been studied before. The model can also take a question type because for the same topic, different types of questions can be asked (see section 5.3).

- It proposes a novel neural network model to solve the problem. A pre-decode mechanism is also explored to improve the model performance.

- The proposed model can be directly trained using a normal question-answering corpus without requiring additional labeling of topics in each input sentence.

- The proposed model is evaluated using the Amazon community question-answering corpus. Experimental results show that our model is effective.

## 2 BACKGROUND AND RELATED WORKS

Our work is inspired by three lines of research: sequence to sequence models, question pattern prediction and question topic selection, multi-source sequence-to-sequence learning.

**Sequence to sequence models** have been successfully applied to many Natural Language Processing tasks especially Neural Machine Translation (Bahdanau et al., 2015; Sutskever et al., 2014) and Sequence Tagging (Liu et al., 2017), etc., Du et al. (2017) tackled question generation using a conditional neural language model with a global attention mechanism and it outperformed a state-of-the-art rule-based system. Their model is a typical sequence to sequence structure using a bidirectional LSTM as the encoder to encode a sentence and a LSTM as the decoder to generate the target question. This structure does not rely on hand-crafted rules or a sophisticated NLP pipeline and can generate more fluent and grammatically correct questions. However, this approach is not targeted and often generate poorly focused questions.

**Question pattern prediction and question topic selection** are two important components of the question generation engine proposed by Duan et al. (2017). In their approach, a question is formed by filling an automatically selected phrase $\mathcal{Q}_t$ into the pattern predicted from pre-mined patterns, which are not done using deep learning. This work illustrates one important hypothesis that it is reasonable to divide one question into two independent parts, i.e., question pattern and question topic. Inspired by this, to control the question type and content in the generation process, we introduce two extra information into the sequence to sequence learning framework named respectively *Topic* (T) and *Question Type* (QT).

**Multi-source sequence-to-sequence learning** aims to integrate information from multiple sources to boost learning. Zoph & Knight (2016) built a multi-source machine translation model and reported up to +4.8 Bleu increases on top of a very strong attention-based neural translation model. Zhou et al. (2017) proposed a neural system combination framework leveraging multi-source NMT, which takes as input the outputs of NMT and SMT systems and produces the final translation. The multi-source framework achieved significant improvement over the best single system output. Inspired by this, we design independent encoders for question type, topic and answer, then a similar mechanism is used to integrate these pieces of information into decoder.

## 3 PROBLEM FORMULATION

We now define and formulate the task of *topic-based question generation*. Given an input sentence **a**, a topic **tc** and a question type **qt** (we use TQT to denote topic and question type together, use **z** to denote (**qt**, **tc**)), the goal is to generate a word sequence **q**, a question of arbitrary length, related to the TQT and the sentence. **qt** is the vector representation of the question type, **tc** is the given topic and is represented as a sequence of tokens $[tc_1, ..., tc_m]$. The topic-based question generation task

is defined as finding the best question $\bar{\mathbf{q}}$ that maximizes the conditional likelihood given $\mathbf{a}$ and $\mathbf{z}$:

$$\bar{\mathbf{q}} = \arg\max_{\mathbf{q}} \log P(\mathbf{q}|\mathbf{a}, \mathbf{z}) = \arg\max_{\mathbf{q}} \sum_{i=1}^{|\mathbf{q}|} \log P(q_i|\mathbf{a}, \mathbf{z}, q_{<i}) \quad (1)$$

where $P(q|a)$ is modeled with a global attention mechanism (Section 4.3).

## 4 MODEL

This section describes the proposed model in detail, which can be trained directly using a question-answering corpus without requiring any additional annotations.

### 4.1 OBSERVATION AND TOPIC EXTRACTION

To generate topic-specific questions, we need the topic in a given answer (or descriptive text) and its question for training. However, training data annotated with topics in the questions and answers are difficult to obtain as it requires a great deal of manual labeling. Here, we propose a simple and effective method to identify topics shared in given question and answer pairs. This method is based on the observation that *the content (or tokens with similar meanings) shared by a given question and answer can be regarded as a topic*. Imagine a multi-round dialogue about tourism shown in the illustration 1, word 'America' is mentioned many times, and it's obvious that this is the topic of the conversation.

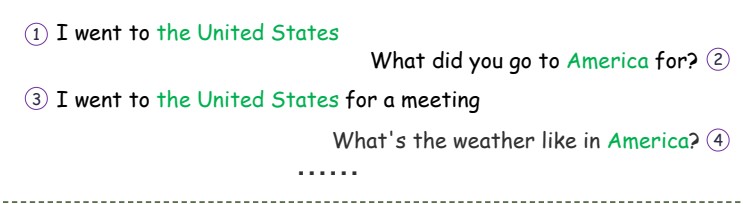

Figure 1: Examples of the observation

Based on this observation, we extract the topic from the given answer $\mathbf{A}$ according to its question $\mathbf{Q}$. $\mathbf{A}$ and $\mathbf{Q}$ are represented as a sequence of tokens $[a_1, ..., a_n]$ and $[q_1, ..., q_m]$ respectively. In the process of topic extraction, we consider all the tokens in $\mathbf{A}$ as candidates for the topic, and consider all the tokens in $\mathbf{Q}$ as voters voting for the candidates similar to $q_i$. Formally, after voting, candidate $a_i$ obtains votes $S_i = \sum_{j=i}^{m} p_{ij} \cdot \text{sim}(a_i, q_j)$, where $p_{ij}$ can be calculated by:

$$p_{ij} = \begin{cases} 1, & \text{if } \text{sim}(a_i, q_j) > \lambda \\ 0, & \text{otherwise} \end{cases} \quad (2)$$

where $\text{sim}(a_i, q_j)$ is the cosine similarity between $a_i$ and $q_j$ calculated using word embeddings of the two words. Candidate $a_i$ is selected as part of the topic $T$ if the average vote $\bar{S}_i = S_i / \sum_{j=1}^{m} p_{ij}$ is greater than a threshold $K$.

*Question type* is another piece of information used in generating a topic-specific question. For example, if one wants a specific type of answer, he/she needs to ask a specific type of question. According to commonly used questioning methods, we divide question types into 7 categories, namely 'yes/no','what','who','how','where','what', and 'others' (very infrequent). Many other types can be grouped into the existing types, e.g., 'whose' belongs to the category 'who'. Since the question type can be fairly easily determined in a given question, we simply employ some keywords to categorize them, which achieves good results.

### 4.2 GENERALIZATION CAPABILITY

Section 4.1 gave a method to extract topics from a raw question-answering training corpus. Given a model learned from the corpus and a topic from those existing topics in the corpus, it is easy

to see that we can generate reasonable questions. However, the problem is whether it is possible to generate a reasonable question given any topics, including those that have not appeared in the training corpus. Under certain conditions, the answer is yes. We explain this here.

Formally, the problem described above is whether our model can generate meaningful question $q_{ij}$ when given an answer $a_i$ and an arbitrary but related tqt (topic and question type) $z_j$ (to some of those tqts in the training corpus). Here we first describe our question generation model and then use the model to answer the question. In fact, the question generation task defined in section 3 can be written as a product of conditional probabilities in various ways. This paper formulates the model in the following way:

$$P(\mathbf{q}|\mathbf{a}, \mathbf{z}) = \sum_{i,j} P(\mathbf{q}|\widetilde{\mathbf{a}}_i, \widetilde{\mathbf{z}}_j) \cdot P(\widetilde{\mathbf{a}}_i|\mathbf{a}) \cdot P(\widetilde{\mathbf{z}}_j|\mathbf{z}) \tag{3}$$

where $\widetilde{\mathbf{a}}_i$ and $\widetilde{\mathbf{z}}_j$ are the internal vector representations for the given answer sentence and topic respectively. We then use these two internal representations to generate the final question. Sutskever et al. (2014) showed that the sentence with the similar meaning will be clustered together by clustering their LSTM hidden states (corresponding to our internal representations). This indicates that similar sentences have similar internal representations, or it can be assumed that they are sampled from the same distribution.

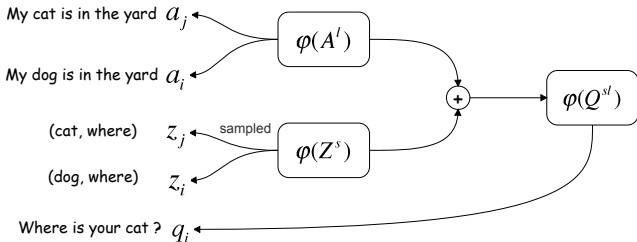

Figure 2: Illustration for Distribution Mapping

As illustration 2 shows, our model uses two internal representations (can be assumed as sampled from $\varphi(A^l)$ and $\varphi(Z^s)$ respectively) to generate a question (can be assumed as sampled from $\varphi(Q^{sl})$) which learns a distribution mapping method $P(\varphi(Q^{sl})|\varphi(A^l), \varphi(Z^s))$ from $\varphi(A^l)$ and $\varphi(Z^s)$ to $\varphi(Q^{sl})$. When given an answer $a_i$ and a tqt $z_i$ which never appeared as a pair in the training data, benefited from the distribution mapping, our model can generate a question similar to $q_j$ because $a_i$ and $a_j$ are from the same distribution $\varphi(A^l)$ and, $z_i$ and $z_j$ are from the same distribution $\varphi(Z^s)$. However, this method requires enough training data to learn each distribution and the mapping relationship between them.

### 4.3 TOPIC-BASED QUESTION GENERATION

To generate topic-specific questions, our model takes two pieces of information, namely a topic and question type pair and an answer. Figure 3 illustrates the overall framework of our model, which has three major components: TQT Encoder, Answer Encoder and Decoder. This framework contains a pre-decode mechanism. We get our basic model if we use $f_t$ marked in the figure to predict words. Below, we describe our model in detail.

**Notation Description**: To simplify the framework in the figure, we omit some connecting lines and attentions, but we use words or symbols to describe those omitted connections and attentions, e.g., word 'attention2' represents an attention similar to 'attention1'.

*Full Connection*: All the 'full connection' components in Figure 3 have the same structure (but different parameters) which can be formulated as:

$$FC(\mathbf{I}) = \mathrm{sigmoid}(\mathbf{W}_s \tanh(\mathbf{W}_t \mathbf{I})) \tag{4}$$

where $\mathbf{I}$ is the input of the 'full connection' component, and $\mathbf{W}_s$ and $\mathbf{W}_t$ are parameters to be learned.

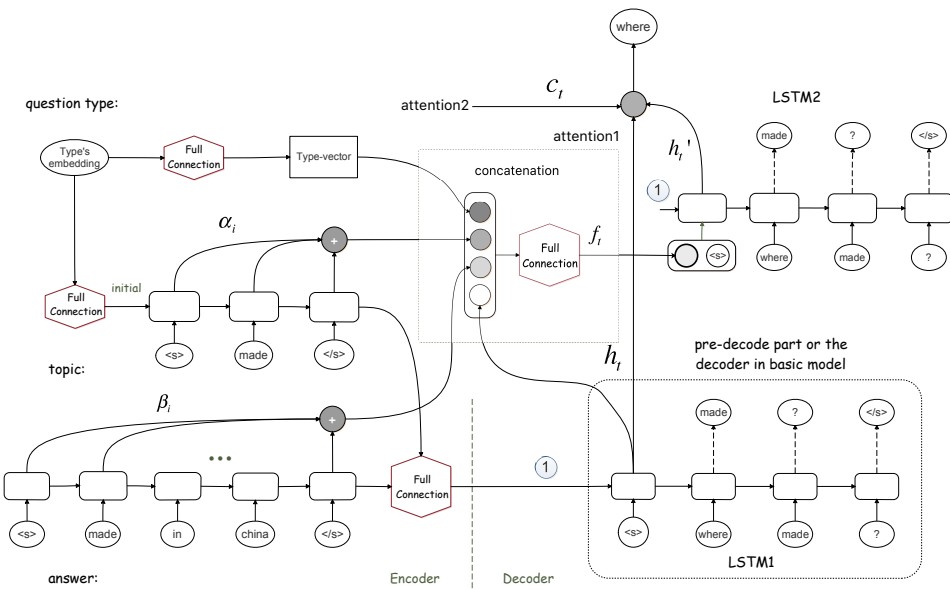

Figure 3: Architecture of the Topic-Specific Question Generation Model

**TQT Encoder**: We also call it topic encoder. It encodes the topic information into an internal vector representation. For question types in tqt (topic and question type), we divide them into 7 categories as we discussed previously, and each category is represented by a type embedding $\mathbf{e\_t}_i$ which can be learned in the training process. The question type encoder uses a full connection function to transduce $\mathbf{e\_t}_i$ to an internal vector representation $\mathbf{qt}_i = FC_1(\mathbf{e\_t}_i)$. We use a bidirectional LSTM to encode the topic,

$$\overrightarrow{\mathbf{b}_t} = \overrightarrow{LSTM}(\mathbf{tc}_t, \overrightarrow{\mathbf{b}_{t-1}})$$
$$\overleftarrow{\mathbf{b}_t} = \overleftarrow{LSTM}(\mathbf{tc}_t, \overleftarrow{\mathbf{b}_{t-1}})$$

(5)

where $\overrightarrow{\mathbf{b}_t}$ is the hidden state at time step $t$ for the forward pass LSTM and $\overleftarrow{\mathbf{b}_t}$ for the backward pass. We use their concatenation, i.e., $[\overrightarrow{\mathbf{b}_t}; \overleftarrow{\mathbf{b}_t}]$, as the hidden state $\mathbf{b}_t$ at time step $t$. As for time step 0, we use $FC_2(\mathbf{e\_t}_i)$ to initialize the LSTM hidden state since we have discussed in section 2 that question type as the first part in a sentence is appropriate for starting a sentence.

**Answer Encoder**: To encoder the answer information, we use another bidirectional LSTM. The formula is similar to equation 5. Denoting the hidden state of this bidirectional LSTM at time step $t$ as $\overrightarrow{\mathbf{ae}_t}$ and $\overleftarrow{\mathbf{ae}_t}$ for the forward and backward directions respectively, we use the concatenation of them as the hidden state $\mathbf{ae}_t$ ($= [\overrightarrow{\mathbf{ae}_t}; \overleftarrow{\mathbf{ae}_t}]$) at time $t$.

We use both the TQT encoder's and answer encoder's outputs for initialization of the decoder hidden state. Firstly, we concatenate each encoder's last hidden state of the forward and backward pass, named respectively $\mathbf{s}_c = [\overrightarrow{\mathbf{b}_{|tc|}}; \overleftarrow{\mathbf{b}_{|1|}}]$ and $\mathbf{s}_a = [\overrightarrow{\mathbf{ae}_{|aw|}}; \overleftarrow{\mathbf{ae}_{|1|}}]$, where $aw$ is a sequence of answer tokens. Then, we take $\mathbf{s}_c$ and $\mathbf{s}_a$ as the input of a full connection function to obtain the initialization of the decoder hidden state:

$$\mathbf{s}_i = FC_2([\mathbf{s}_c; \mathbf{s}_a])$$

**Decoder**: Since we use a LSTM to decode the internal representation and generate questions, in our model, the conditional likelihood $P(q_t|\mathbf{a}, \mathbf{z}, q_{<t})$ in equation 1 can be calculated by:

$$P(q_t|\mathbf{a}, \mathbf{z}, q_{<t}) = \text{softmax}(\mathbf{f}_t)$$
$$\mathbf{f}_t = FC_3([\mathbf{qt}_t; \mathbf{tc}_t; \mathbf{ac}_t; \mathbf{h}_t])$$

(6)

where $\mathbf{h}_t$ is the hidden state of the decoder LSTM; $\mathbf{qt}_t$ is the question type encoder's output; $\mathbf{tc}_t$ and $\mathbf{ac}_t$ have the same calculation method. Here we only give the calculation formula for $\mathbf{tc}_t$:

$$\mathbf{tc}_t = \sum_{i=1}^{m} \alpha_i \mathbf{b}_i$$

(7)

where $\mathbf{b}_i$ is the hidden state at time step $i$ for the bidirectional LSTM in topic encoder, and its weight $\alpha_i$ is calculated by:

$$\alpha_i = \frac{exp(e_i)}{\sum_{i=1}^{m} exp(e_i)} \tag{8}$$

where $e_i = \mathbf{h}_t \mathbf{W}_a \mathbf{b}_i$ scores how well $\mathbf{h}_t$ and $\mathbf{b}_i$ match.

**Pre-decode Mechanism**: The topic extraction approach introduced in section 4.1 is simple and effective. However, the topic extracted by this approach still mixed with some noise. To improve the model performance and filter out some noise, we explore a pre-decode mechanism. The main idea of this mechanism is to use another LSTM to obtain a pre-decode result $\mathbf{f}_t$ at time step $t$ as a filter to influence the final prediction. In the pre-decode method, the conditional likelihood $P(q_t|\mathbf{a}, \mathbf{z}, q_{<t})$ formula in equation 6 needs to be rewritten as:

$$P(q_t|\mathbf{a}, \mathbf{z}, q_{<t}) = \mathrm{softmax}(\mathbf{W}_s tanh(\mathbf{W}_l[\mathbf{c}_t; \mathbf{h}_t; \mathbf{h}'_t])) \tag{9}$$

where $h_t$ is the hidden state of the pre-decoder LSTM; $h'_t$ is the hidden state of the final decoder LSTM; $\mathbf{c_t} = FC_4([\mathbf{qt}_t; \mathbf{tc}'_t; \mathbf{ac}'_t])$. The weighted sum $\mathbf{tc}_t$ in equation 7 needs to add the weight (or filter) $\gamma_i$ calculated by the pre-decode result, denoted as $\mathbf{tc}'_t$:

$$\mathbf{tc}'_t = \sum_{i=1}^{m} \alpha_i \gamma_i \mathbf{b}_i \tag{10}$$

where

$$\gamma_i = \frac{exp(e'_i)}{\sum_{i=1}^{m} exp(e'_i)} \tag{11}$$

where $e'_i = \mathbf{f}_t \mathbf{W}_a \mathbf{b}_i$; $\alpha_i$ has the same calculation method as that for equation 8 except $e_i = \mathbf{h}'_t \mathbf{W}_a \mathbf{b}_i$. The hidden state of the final decoder LSTM $\mathbf{h}'_t$ is obtained by combining the previous hidden state $\mathbf{h}'_{t-1}$, the word representations $\mathbf{em}_t$ at the current time step, and the pre-decode result $\mathbf{f}_t$:

$$\mathbf{h}'_t = LSTM([\mathbf{f}_t; \mathbf{em}_t], \mathbf{h}_{t-1}) \tag{12}$$

## 5 EXPERIMENT

We now evaluate the proposed model for our topic-based question generation task. As there is no existing method that can perform this new task, we compare with conventional question generation (which our model can do as well) without any given topic or question type.

### 5.1 DATA PREPARATION

We use the Amazon question/answer corpus[1] (AQAD) (Wan & McAuley, 2016) for training and testing. The dataset contains questions and answers about products and services from Amazon. It has around 1.4 million answered questions.

For pre-processing, we remove those question-answer pairs (using simple patterns) with answers that are impossible to be used to generate meaningful questions, like "yes/no", "yes, you can" and "no, you cannot," etc. Since we focus on generation of a single question from one answer, we also remove those question-answer pairs with multiple questions or the question length with more than 50 words. We are finally left with about 900k question-answer pairs for our experiments.

We use the approach proposed in section 4.1 to extract topics and decide the question type. Pre-trained word embedding[2] (Pennington et al., 2014) is employed to calculate the similarity in equation 2. We set the topic selection threshold $K$ to 0.4, and the word similarity threshold $\lambda$ to 0.3 (section 4.1). For each question/answer pair, we select no more than 5 words as the topic.

We randomly divide the corpus into a training set, a development set (with about one thousand q/a pairs), and a test set (with about one thousand q/a pairs). We report the results on the test set.

---

[1]http://jmcauley.ucsd.edu/data/amazon/qa/
[2]We use glove.840B.300d.

## 5.2 Training Details

The hyper-parameters in our system are described as follows. We share the word embedding between the answer and the question. We limit the shared vocabulary to 30k in our experiments. The number of hidden units is 600 and the word embedding dimension is 300. We set the number of layers of LSTM to 1 in both encoder and decoder. The network parameters are updated with the Adam algorithm (Kingma & Ba, 2014) with learning rate 0.0001.

For the experiment of our system in the conventional question generation setting, since there is no topic given at the test time, we employ a sentence classification model (Kim, 2014) to predict the question type from the given answer and a sequence labeling model (Liu et al., 2017) to extract topics from the given answer. Both models are trained using automatically extracted labels.

## 5.3 Results and Analysis

**Automatic Evaluation**: Following Du et al. (2017), we use the evaluation package released by Chen et al. (2015). The package includes BLEU 1 to 4, METEOR and ROUGE$_L$.

*Baseline*: We use the latest model from (Du et al., 2017) as our baseline, which is a conventional question generation system using a sequence-to-sequence model. It significantly outperforms state-of-the-art rule-based systems.

| Model | BLEU 1 | BLEU 2 | BLEU 3 | BLEU 4 | METEOR | ROUGE$_L$ |
|---|---|---|---|---|---|---|
| Du et al. | 33.30 | 19.69 | 13.19 | 9.25 | 13.68 | 35.68 |
| Ours (a-qt a-tc) | 33.29 | 19.00 | 12.11 | 8.09 | 13.06 | 34.66 |
| Ours* (a-qt a-tc) | 34.49 | 19.95 | 13.09 | 9.09 | 13.85 | 36.10 |
| Ours* (a-qt g-tc) | 35.87 | 20.76 | 13.57 | 9.51 | 14.57 | 35.92 |
| Ours* (g-qt a-tc) | 37.20 | 21.97 | 14.61 | 10.16 | 14.58 | 37.40 |
| Ours* (g-qt g-tc) | 42.30 | 26.23 | 18.16 | 13.09 | 17.50 | 42.48 |

Table 1: Experimental results for different models based on automated evaluation. The symbol string 'a-qy g-tc' in the table means using automatically generated question type and given topic to test. Specifically, 'a' is for *automatic*, 'g' is for *given* (question type or topic), 'qt' is for *question type*, and 'tc' is for *topic*. Ours* is our model with the pre-decode mechanism.

Table 1 shows the results. The first three rows are for conventional question generation, and the last three rows are for different variations of the proposal topic-based question generation. From the table, we can make the following observations:

(1) Based the results from the first three rows, we can see that although our model is not designed for conventional question generation, with the pre-decode mechanism it outperforms the state-of-the-art baseline in four out of 6 test metrics. Our model (row 3) did not achieve even better results due to the difficulty of automated sentence classification and topic labeling. Table 2 shows their results.

(2) Based on all the results, we can see that our models (our*) with the pre-decode mechanism significantly outperforms our basic model (row 2) which shows the validity of pre-decode mechanism.

(3) When the topic is given, regardless whether the question type is given or not, our models (last three rows) outperform the baseline (row 1). When the question type is also given, the results are markedly better. Although these results are expected as our models are given more information, in practical applications, one seldom wants a random question. A question should be generated on a topic that is related to the current conversation and with a desired type of answer.

| Model | Accuracy | Model | Recall | Presion | F1-score |
|---|---|---|---|---|---|
| Sentence Classification | 57.58 | Sequence Labeling | 50.24 | 52.67 | 55.36 |

Table 2: Test results for sentence classification and sequence labeling.

**Human Evaluation**: We also perform human evaluation to measure the quality of questions generated by our system and the baseline. We consider naturalness modality, which indicates the grammaticality, fluency and rationality. We randomly sampled 100 sentence-question pairs and completely shuffled the order of tested systems. We ask three Ph.D.students to rate the pairs in terms of the modality on the 1 to 5 scale (5 for the best and the total score is scaled to the full score of 100).

| | Du et al. | Ours* | | | |
|---|---|---|---|---|---|
| | | a-qt a-tc | a-qt g-tc | g-qt a-tc | g-qt g-tc |
| Score | 66.47 | 65.07 | 67.4 | 66.27 | **68** |

Table 3: Experimental results for different models based on human evaluation. The symbol string a-qt g-tc in the table has the same meaning as in Table 1. The perfect score is 100.

Table 3 shows that our model gets the highest score on the condition of 'a-qt g-tc' and 'g-qt g-tc', which shows that our model generates more natural questions (i.e. grammaticality, fluency and rationality). Our model (columns 2 and 4) did not achieve even better results due to the wrong topic and we'll explain this with examples in the *Case Study* section.

**Case Study**: In Table 4, we present some sample questions generated by our model and the baseline. From the first sample, we can see that the question generated by our model is more consistent with the given sentence. By comparing our model output of different topics, we can find that our model can still generate a high quality question when given an inaccurate topic. This means that our model has a certain anti-interference ability.

In the second sample, the baseline model generates a question that is irrelevant to the given sentence. However, our model can generate question relevant to the given topic and sentence. In this sample, our model generates an irrelevant question due to the wrong given topic 'tracks' and generate a natural and relevant question when given a reasonable topic. It is for this reason that our model (when taking 'a-qt a-tc' as input) did not achieve even better results in human and automatic evaluations.

| | |
|---|---|
| **Sentence 1**: they hold 8 ounces, 1 cup.
**Human:**: how many ounces do they hold?
**Du et al.**: how much do they hold?
**Our* (a-qt a-tc)**: how many ounces do these hold? (given topic: *hold*)
**Our* (g-qt g-tc)**: how many ounces does each hold? (given topic: *hold 8 ounces*) | **Sentence 2**: you can forward and reverse tracks by holding down the up or down volume button for a few seconds.
**Human:**: can you forward and reverse music with a button on the speaker?
**Du et al.**: how do you turn it on/off?
**Our***: can you adjust the volume of the track? (given topic: *tracks*)
**Our***: does this come with a reverse forward button? (given topic: *forward reverse button*) |

Table 4: Sample output questions generated by human, our system, and Du et al's system.

**Topic Effect Analysis**: In section 4.2, we discussed the generalization capability of our model and gave a theoretical explanation. Here, we use the experimental results to support the argument we made in section 4.2.

| **Sentence:** | | *bottle says "made in usa".* |
|---|---|---|
| **Human:** | | *where is this product made?* |
| **Topic** | **Question Type** | **Generated Question** |
| made | yes/no | is this product made in the usa ? |
| made | what | what is the country of origin ? |
| made | where | where is this product made ? |
| bottle | what | what is the manufacturing date of the bottle ? |
| bottle | where | where is the bottle of origin ? |
| says | who | who is the manufacturer ? |

Table 5: Questions generated from our model by given arbitrary question type or topic.

Table 5 shows several questions generated from a given sentence according to different question types and topics. The first three rows are for given the same topic but different question types. The results show that our model can change the questioning method according to the given question type. Note that some question types, i.e. 'yes/no' and 'what' for the given sentence have not appeared in the training corpus. The last three rows are for given arbitrary topics. In this situation, questions generated by our model are still rational and consistent with the given topic, question type and the sentence.

## 6 CONCLUSIONS

In this paper, we proposed the new task of topic-based question generation, which has not been investigated before. Based on our experiences, we believe this is a more useful question generation setting than the conventional setting without a given topic because a question without the knowledge of its topic is of limited use in actual conversations. We then proposed a novel network architecture to perform the task, and discussed its generalization ability. Experimental results showed that the proposed model performed slightly better than the state-of-the-art baseline in the conventional setting (although our model not designed for this setting), and performed markedly better in the proposed new setting.

ACKNOWLEDGMENTS

Use unnumbered third level headings for the acknowledgments. All acknowledgments, including those to funding agencies, go at the end of the paper.

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
