# OpenReview forum: "Topic-Based Question Generation"
_ICLR.cc/2018/Conference — Invite to Workshop Track_

### Official Review · AnonReviewer3 · 2017-11-26
**Motivation, experiments, and evaluation are flawed**

**Rating:** 3
**Confidence:** 5

**Review:**

This paper presents a neural network-based approach to generate topic-specific questions with the motivation that topical questions are more meaningful in practical applications like real-world conversations. Experiments and evaluation have been conducted on the AQAD corpus to show the effectiveness of the approach.

Although the main contributions are clear, the paper contains numerous typos, grammatical errors, incomplete sentences, and a lot of discrepancies between text, notations, and figures making it ambiguous and difficult to follow.

Authors claim to generate topic-specific questions, however, the dataset choice, experiments, and examples show that the generated questions are essentially keyword/key phrase-based. This is also apparent in Section 4.1 where authors present some observation without any supporting proof or empirical evidence. Moreover, the example in Figure 1 shows a conversation, but, typically, in an ongoing multi-round conversation people do not tend to repeat the keywords or key phrases or named entities, and topic shifts might occur at any time.

Overall, a misconception about topic vs. keywords might have led the authors to claim that their work is the first to generate topic-specific questions whereas this has been studied before by Chali & Hasan (2015) in a non-neural setting. "Topic" in general has a broader meaning, I would suggest authors to see this to get an idea about what topic entails to in a conversational setting: https://developer.amazon.com/alexaprize/contest-rules . I think the proposed work is mostly related to: 1) "Towards Natural Question-Guided Search" by Kotov and Zhai (2010), and 2) "K2Q: Generating Natural Language Questions from Keywords with User Refinements" by Zheng et al. (2011), and other recent factoid question generation papers where questions are generated from a given fact (e.g. "Generating Factoid Questions With Recurrent Neural Networks: The 30M Factoid Question-Answer Corpus" by Serban et al. (2016)).

It is not clear how the question types are extracted from the given sentences. Please provide details. Which keywords are employed to accomplish this? Also, please explain the absence of the "why" type question.

Figure 3 and the associated descriptions are very hard to follow. Please draw the figure by matching it with the descriptions.  Where are the bi-LSTMs in the figure? What are ac_t and em_t?

My major concern is with the experiments and evaluation. The dataset essentially contains questions about product reviews and does not match authors motivation/observation about real-world conversations. Moreover, evaluation has been conducted on a very small test set (just about 1% of the selected corpus), making the results unconvincing. More details are necessary about how exactly Kim's and Liu's models are used to get question types and topics.

Human evaluation results per category would have been more useful. How did you combine the scores of the human evaluation categories? Also, automatic evaluation and human evaluation results do not correlate well. Please explain.

---

> ### Author Response · Authors · 2017-12-20
> **Thank you for your review and comments**
>
> 1. We are very sorry about the typos, grammatical errors, etc. We will fix them in the final version. And we will fix the incomplete Figure 3 in the new version.
>
> 2. Thank you for pointing out the "topic" problem. The terms topic and keyword are fairly ambiguous. We can use the keyword in the new version. Our work is quite different from the paper that you mentioned. Our is not generating questions totally based on given keywords. Our motivation is based on the fact that several questions can be asked based on a given sentence. Hence, we want to generate questions about the given "subject" or "theme" conditioned on the given descriptive text.
>
> 3. We simply use keywords such as "what","how", etc. to extract question types from the questions in the training data set. We will add details in the new version. The absence of the “why” type is a mistake, we missed it. In that case, all the "why" data is divided into the "other" type. Especially, the experimental results for other types won't be changed. And our conclusion still holds.
>
> 4. It is impossible for people to ask questions about some things but don't mention them in the question. For example: **question** "are the ==tips== interchangeable ?", **answer**: "the ==tips== are one piece metallic construction solid glued in place .". So our observation/motivation is still true. As for the size of the test set, we believe that thousands of samples are enough to test the model since those samples are randomly sampled from a big dataset. For example, the NIST test set for machine translation consists of thousands samples too and its training set can be 2 million sentence pairs (we can see that in lots of machine translation researches). But we will employ a larger dataset in the next experiment. We will add more details about Kim's and Liu's models in the revised version.
>
> 5. For "How did you combine the scores of the human evaluation categories?",  that is a problem. We also found it is hard to combine them, so we ask the participants to give one score according to the naturalness modality from an overall view. For "why automatic evaluation and human evaluation results do not correlate well," in general, e.g. "+how tall is+ the =lamp= itself ?", regarding different ways to formulate questions, it needs many words (marked by ++) to express ,  while regarding to what subject/topic  (marked by ==) to ask questions, it needs just one or two. Since BLEU favours longer references while humans judge based on overall expressions, that's the reason for different performances of automatic BLEU evaluation and human judgments.

---

### Official Review · AnonReviewer1 · 2017-11-27
**Review from AnonReviewer1**

**Rating:** 4
**Confidence:** 4

**Review:**

This paper proposed a topic-based question generation method, which requires the input of target topic in addition to the descriptive text. In the proposed method, the authors first extract the topic based on the similarity of the target question token and answer token using word embedding. Then, the author proposed a topic-specific question generation model by encoding the extracted topic using LSTM and a pre-decode technique that the second decoding is conditioned on the hidden representation of the first decoding result. The authors performed the experiment on AQAD dataset, and show their performance achieve state-of-the-art result when using automatically generated topic, and perform better when using the ground truth topic.

[Strenghts]

This paper introduced a topic-based question generation model, which generate question conditioned on the topic and question type. The authors proposed heuristic method to extract the topic and question type without further annotation. The proposed model can generate question with respect to different topic and pre-decode seems a useful trick.

[Weaknesses]

This paper proposed an interesting and intuitive question generation model. However, there are several weaknesses existed:

1: It's true that given a descriptive text, it is often possible to ask many types of questions. But it also leads to different answers. In this paper, the authors treat the descriptive text as answers, is this motivation still true if the question generation is conditioned on answers, not descriptive text? Table 4 shows some examples, given the sentence, even conditioned on different topics, the generated question is similar.

2: In terms of the experiment,  the authors use AQAD to evaluate proposed method. When the ground truth topic is provided, it's not fair to compare with the previous method, since knowing the similar word present in the answer will have great benefits to question generation.

If we only consider the automatically generated topic, the performance of the proposed model is similar to the previous method (Du et al). Without the pre-decode technique, the performance is even worse.

3: In section 4.2, the authors claim this is the theoretical explanation of the generalization capability of the proposed model (also appear in topic effect analysis). It is true that the proposed method may have better compositionality, but I didn't see any **theoretical** explantation about this.

4: The automatically extracted topic can be very noisy, but the paper didn't mention any of the extracted topics on AQAD dataset.

[Summary]

a topic-based question generation method, which requires the input of target topic in addition to the descriptive text. However, as I pointed out above, there are several weaknesses in the paper. Taking all these into account, I think this paper still needs more works to make it solid and comprehensive before being accepted.
Add Comment

---

> ### Author Response · Authors · 2017-12-20
> **Thank you for your review and comments**
>
> 1. We believe our motivation still true even if the question generation is conditioned on answers. That is because: (1) quite many answers can’t be classified properly even by people. Please see the example for reviewer 1. (2) If there is a good correspondence among all the questions and answers, the accuracy of the sentence classification and labeling will not be so low (see Table 2 in the paper). (3) Actually ,  daily conversation can also be regarded as a kind of inquiry and answer. (4) Here, we given an example to explain that our data set still match our motivation/observation : **question**: "are the =tips= interchangeable ?", **answer**: "the =tips= are one piece metallic construction solid glued in place .". On the other hand, we believe the generated questions conditioned on different topics on Table 4 are not similar. The first question in the last three rows is about "the manufacturing date", and the second question is asking "the origin of the bottle", while the third one is about the "manufacturer". Of course, it is right to say they are similar if we talk about the similarity at a higher level since they are all about "manufacture". And this is because they have the same context "bottle says 'made in usa'."
>
> 2. Yes, we know and that is also the reason why we split the Table 1 into 2 parts. Let us only consider the automatically generated topic. It is true that our performance is even worse without the pre-decode technique (but we have pre-decode technique). (1) We give the reason for that in the paper. It is because we would like to build a system to generate controlled questions but the poor sentence classification and labeling accuracy lead our system to generate wrong sentences. (2) As there is no existing method that can perform our proposed new task, we compare with the conventional question generation (our model is not designed for that purpose). (3) The inconsistency of training and testing puts our model at a disadvantage. To achieve the proposed goal, we have to use the extracted ground truth to train, but to have a fair comparison with the conventional question generation method we then must test our model using auto-generated topics and question types.
>
> 3. Thank you for pointing out this problem. We will correct that. What we have given is not a strict mathematical proof, it is a brief explanation. As assumed it is hard to give a mathematical proof in neural networks.
>
> 4. We used several methods to tackle the noise problem. (1) We removed the stop words using NLTK(a natural language toolkit). (2) We employed the Bagging approach in the extraction process. (3) We proposed the pre-decode technique.  At the same time we allow the topic to correspond to the empty value. According to our statistics, about 32.6% of the sentences in the training set failed to extract the topic. We would rather have it corresponding to the empty value than to introduce noise. Here, we given some examples:
> (1) **answer**  : "yes it is . it is a great product . it works with all devices except samsung 7 inch tablet . it is not the keyboard it is samsung . good luck."  **question** :  "is this keyboard compatable with the acer a500 tablet ?" **topics** : "tablet keyboard"
> (2) **answer** : "yes . it is comfortable even with my glasses on ."  **question** : "are these comfortable if you wear glasses ? do they hurt your ears from physical contact ?" **topics** :  "comfortable glasses"
> (3) **answer** : "it does . but it is not as good as smart phone apps available today ." **question** : "this equipment translates from spanish to portuguese ?" **topics** : "" (empty)

---

### Official Review · AnonReviewer2 · 2017-11-27
**Interesting paper on a question generation approach focusing on some topics and question type with promising experimental results.**

**Rating:** 8
**Confidence:** 3

**Review:**

The authors propose a scheme to generate questions based on some answer sentences, topics and question types. Topics are extracted from questions using similar words in question-answer pairs. It is similar to what we find in some Q&A systems (like lexical answer types in Watson). A sequence classifier is also used to tag the presence of topic words. Question types correspond mostly to salient questions words. LSTMs are used to encode the various inputs and generate the questions.

The paper is well written and easy to follow. I would expect more explanations why sentence classification and labeling results presented in Table 2 are so low.

Experimental results on question generation are convincing and clearly indicate that the approach is effective to generate relevant and well-structured short questions.

The main weakness of the paper is the selected set of question types that seems to be a fuzzy combination of answer types and question types (for ex. yes/no). Some questions type can be highly ambiguous; for instance “What” might lead to a definition, a quantity, some named entities... Hence I suggest you revise your qt set.

I would also suggest, for your next experiments, that you try to generate questions leading to answers with list of values.

---

> ### Author Response · Authors · 2017-12-20
> **Thank you for your helpful feedback!**
>
> We are also surprised about the poor accuracy of sentence classification and labeling. Based on observation and analysis of the result. We noticed that the difficulty lies in the fact that multiple questions can be asked based on one given sentence. It is also based on this fact that we proposed to ask questions targeted toward some relevant indicators.
>
> For example, for a given answer "it's sort of got a cardboard feel to it , but it feels very sturdy nonetheless." , the question might be "how does it feel?", "what does it feel like?" or "is it sturdy enough?" but the ground truth question is "what material is it made out of ?".  So, it is quite challenging to generate questions consistent with the given ground truth based only on the given answer.
>
> We will revise the qt set. Thanks. There are actual ambiguities existing in some question types. Since we want to propose a scheme to generate controlled questions in an unsupervised manner, so there is little information can help us to identify the question types in detail. But the "answer types" you mentioned greatly inspired us. We can do that by considering the answers type. Thanks.
>
> Thanks for experiment suggestion. It's an interesting idea. We will do that in the next experiment. We think an special memory mechanism can be designed to do that.

---

### Decision · Program_Chairs · 2018-01-29
**ICLR 2018 Conference Acceptance Decision**

**Decision:**

Invite to Workshop Track

**Comment:**

The pros and cons of the paper under consideration can be summarized below:

Pros:
* Reviewers thought the underlying model is interesting and intuitive
* Main contributions are clear

Cons:
* There is confusion between keywords and topics, which is leading to a somewhat confused explanation and lack of clear comparison with previous work. Because of this, it is hard to tell whether the proposed approach is clearly better than the state of the art.
* Typos and grammatical errors are numerous

As the authors noted, the concerns about the small dataset are not necessarily warranted, but I would encourage the authors to measure the statistical significance of differences in results, which would help alleviate these concerns.

An additional comment: it might be worth noting the connections to query-based or aspect-based summarization, which also have a similar goal of performing generation based on specific aspects of the content.

Overall, the quality of the paper as-is seems to be somewhat below the standards of ICLR (although perhaps on the borderline), but the idea itself is novel and results are good. I am not recommending it for acceptance to the main conference, but it may be an appropriate contribution for the workshop track.